# Ingroup favoritism and outgroup derogation in intergenerational cooperation

Hirotaka Imada [1,2] ✉, Yukako Inoue[3], Alice Yamamoto-Wilson[4], Tatsuyoshi Saijo[5,6] & Nobuhiro Mifune[2,7]

Issues related to sustainability (e.g., climate change and over-fishing) often manifest themselves as intergenerational social dilemmas, where people are faced with a choice between self-serving, unsustainable behavior and sustainable, personally costly behavior. Extending the previous literature on (non-intergenerational) intergroup cooperation, we tested whether group membership of the future generations influenced sustainable decision-making. In two preregistered studies using the intergenerational sustainability dilemma game, we found that individuals were more likely to make a sustainable (vs. selfish) decision when they believed that their current behavior would benefit future ingroup members, whereas more selfish decisions were made when benefits would accrue to outgroup members. These findings held in both the minimal group (Study 1: N = 1393) and national group (Study 2: Japan vs. China, N = 1781) contexts. The effect of ingroup intergenerational membership on cooperation was mediated by higher felt responsibility for future generations in both minimal and national group contexts. The effect of outgroup membership on intergenerational cooperation was mediated by a reduced sense of reputational concern in the minimal group context and by reduced affinity, legacy motivation, and responsibility for future generations in the nationality context.

Many societal issues manifest themselves as social dilemmas, situations in which personal and collective interests conflict with each other[1,2]. Some sustainability issues have such an underlying structure, and individuals are often faced with a choice between self-benefit maximizing unsustainable behavior and sustainable behavior that benefits future generations (i.e., intergenerational social dilemmas[3]). By intergenerational, we refer to interactions and consequences that extend beyond the currently living population, influencing individuals who do not yet exist. Some issues are global (e.g., climate change), but others are local (e.g., overfishing), which makes the marked psychological difference; in local intergenerational social dilemmas, sustainable behavior is targeted only at future generations of one's own local community. Is it easier for individuals to sacrifice personal gains to exclusively benefit future ingroup members compared to future generations, regardless of group membership? How willing are they to cooperate when their cooperation predominantly influences the future of an outgroup? Addressing these questions helps us understand whether and how we should approach global and local sustainability issues differently.

We thus empirically investigated whether intergenerational cooperation is greater when it solely benefits future generations of one's own group than when it benefits those of an outgroup or unclassified future strangers.

In non-intergenerational social dilemmas, previous studies have robustly documented ingroup favoritism, the tendency to cooperate more for ingroup members than outgroup members[4]. In addition, experimental and meta-analytic evidence suggests that bounded indirect reciprocity plays a crucial role in shaping ingroup favoritism[4–7]. Namely, Yamagishi and colleagues argued that intergroup contexts activate the heuristic that indirect reciprocity, a system in which individuals reward one another with cooperation based on their reputations, is bounded by group membership[5–8]. This heuristic leads people to believe that when they face social dilemmas with ingroup members, but not with outgroup members, their cooperation brings long-term reputational benefits and other ingroup members are willing to cooperate with them[7–10].

Intergenerational social dilemmas are distinct from non-intergenerational ones in that because individuals from different and distant generations do not live in the same time frame, (1) decision-makers (i.e.,

[1]Department of Psychology, Royal Holloway, University of London, Egham, UK. [2]Research Institute for Future Design, Kochi University of Technology, Kochi, Japan. [3]Department of Social Psychology, Yasuda Women's University, Hiroshima, Japan. [4]Independent Researcher, Tokyo, Japan. [5]Kyoto University of Advanced Science, Kyoto, Japan. [6]Research Institute for Humanity and Nature, Kyoto, Japan. [7]School of Economics & Management, Kochi University of Technology, Kochi, Japan. ✉e-mail: Hirotaka.Imada@rhul.ac.uk

those in the present generation) cannot assume their reputation in future generations benefits them and (2) they cannot expect their sustainable behavior to be reciprocated by future generations (i.e., they do not gain from reciprocation by future generations[3]). Thus, the key psychological mechanisms behind ingroup favoritism are structurally absent in intergenerational social dilemmas, implying that intergenerational cooperation would not be influenced by whether or not beneficiaries (i.e., future generations) are ingroup members.

However, recent studies have suggested that present ingroup favoritism would be positively correlated with future ingroup favoritism[11]. Specifically, Aaldering and colleagues[11] demonstrated that the tendency to universally and parochially cooperate with others in the same generation predicts that with others in the future generation. As such, ingroup favoritism that has been robustly observed in the non-intergenerational contexts would emerge in the intergenerational contexts too, albeit to a lesser degree[11]. In addition, previous studies on psychological underpinnings for sustainable, future-oriented behavior suggest that shared group membership between the current and future generations promotes intergenerational cooperation via a set of psychological mechanisms that are unique to intergenerational decision-making[12,13]: affinity[3,13-15], legacy[14,16], and generativity (responsibility)[12,17-19].

Affinity refers to the extent to which individuals feel empathetic towards, can visualize, and understand the consequences of their behavior for future generations[3]. Previous studies demonstrated that increased affinity with future generations promoted intergenerational cooperation[3] and intergenerational solidarity[20]. Wade-Benzoni and Tost[12], in their theoretical review, held that intergenerational identification (i.e., the extent to which individuals feel they share a common group identity with people in other generations[21]) is positively associated with affinity, suggesting that shared group membership with others in future generations increases sustainable behavior via increased affinity. In other words, when individuals are aware that future generations whose behaviors they affect belong to the same group (i.e., the same entity spanning from the current generation to the future generations), they may feel increased identification and affinity and, thus, display increased intergenerational cooperation.

Legacies offer individuals an enduring meaning to their identity[14]. Correspondingly, legacy motivation is defined as the personal motive to be remembered, to leave an enduring meaning to their identity, and to leave something for future generations in order to achieve a feeling of symbolic immortality[12,22,23]. Previous studies found that legacy motivation is positively associated with sustainable future-oriented behavior[13,15,24-26]. Importantly, legacies have the function of extending self into the future and gearing self-interests towards future others' interests. Fox et al.[22] pointed out that people often perceive future others to be those who belong to the same group, and this suggests that legacy motivations may promote intergenerational cooperation toward future ingroup members, but not necessarily for future outgroup members. As such, the positive effect of legacy motivations may be bounded by group membership and contribute to shaping future ingroup favoritism.

We note that there are two distinct types of legacy motivations: the reputation legacy and the impact legacy[16,27]. The former refers to the desire to be remembered as a good person (henceforth "reputational concern") and the latter refers to the desire to leave a positive impact on future generations (henceforth "legacy motivation"). Since the relative impact of the two legacy motivations on intergenerational cooperation is unclear, particularly in intergroup contexts, we measured them separately and explored how they each contributed to intergenerational ingroup favoritism in the present research.

Generativity was first introduced by Erikson[19] in his stages of development theory, and originally defined as the concern about nurturing future generations (e.g., bearing and raising children). McAdams and St. Aubin[12] further expanded the concept and defined it as the configuration of psychological features constellated around the individual and societal goals of providing for the next generations. According to their theory of generativity, it includes the motives, concerns, and actions towards future generations.

Previous studies showed that generativity is related to high proenvironmental and sustainable attitudes and behaviors[17,18]. One of the most important and well-studied tenets of generativity is responsibility[28-30], and it refers to the sense of moral obligation towards future generations[30]. Previous studies revealed that the perception of responsibility promoted proenvironmental, future-oriented attitudes[29-32]. Perceived responsibility corresponds with the perceived demand from one's society[12] and it is conceivable that individuals feel responsible to be generative for their society, i.e., future members of the society. If so, similar to legacy motivation, responsibility promotes intergenerational cooperation, especially towards future ingroup members, contributing to the emergence of ingroup favoritism in intergenerational social dilemmas.

Regarding intergenerational cooperation with future outgroup members, recent studies have suggested that individuals have less moral concern for the welfare of outgroup members in the present generation than those in far future generations[33]. This suggests that individuals may be less willing to cooperate for future outgroup members than with strangers in the future (i.e., future outgroup derogation). We note that previous studies also suggest that individuals prefer benefiting both future ingroup and outgroup members over exclusively benefiting future ingroup members[34]. However, those studies did not directly test whether outgroup membership per se discouraged intergenerational cooperation, and thus, we preregistered to test the prediction that we would observe future outgroup derogation.

To recapitulate, previous studies suggest that the positive effects of affinity, reputational concern, legacy motivation, and responsibility (generativity) may be bounded by group membership, and we examined how group membership of the future generations shapes intergenerational cooperation. In the present research, we conducted two studies to elucidate intergroup intergenerational cooperation in minimal and national group contexts. We used the intergenerational sustainability dilemma game (ISDG[35,36]), which allowed us to directly observe participants making a choice between self-serving, unsustainable behavior and sustainable behavior (i.e., intergenerational cooperation). We also aimed to explore the psychological mechanisms behind the hypothesized future ingroup favoritism and outgroup derogation by measuring affinity, legacy (reputational concern and legacy motivation), and generativity (responsibility).

In Study 1, in order to examine the effect of group membership per se, we used the minimal group paradigm[37,38] in which participants were divided into two arbitrarily created experimental groups. Given the discussions above, we hypothesized that individuals are more and less likely to make a sustainable choice when members of the future generations belong to an ingroup (H1) and an outgroup (H2), respectively. In addition, we explored the influence of affinity, legacy, and perceived responsibility as potential psychological underpinnings of intergroup intergenerational cooperation. We preregistered the study procedure, study material, the hypothesis and hypothesis testing, data exclusion, and target sample size and justification at https://osf.io/qjwa4 (preregistration date: February 19, 2024).

While studies with minimal group contexts offer valuable insights as to how group membership itself guides intergenerational cooperation, they nevertheless lack ecological validity. To gain more practical insights as to how group membership influences intergenerational cooperation, in Study 2, we used national groups (Japan vs. China), instead of minimal groups, as a focal intergroup context. We preregistered the study procedure, study material, the hypothesis and hypothesis testing, data exclusion, and target sample size and justification at https://osf.io/5wdte (preregistration date: March 17, 2024).

All studies were approved by the institutional ethics committee, Kochi University of Technology Research Ethics Review Committee (pilot study) and the University Research Ethics Committee at Royal Holloway, University of London (Studies 1 and 2), prior to data collection.

## Methods
### Study 1
**Pilot study, participants, and design.** The study followed a between-subjects design with three conditions (group: ingroup vs. outgroup vs.

control). We preregistered to test the two hypotheses with a logistic regression with the following dummy coding: Dummy 1: ingroup condition vs. control condition (H1); Dummy 2: outgroup condition vs. control condition (H2). We conducted a priori power analysis using data from a pilot study. We first conducted a pilot study (N = 599, $M_{age}$ = 35.89, SD = 12.77, 294 men, 304 women). The study procedure was identical to Study 1, but the content of the post-experimental questionnaire slightly differed (visit https://osf.io/p2wn9/ for more details). It revealed that 1317 participants would be sufficient to detect the smallest effect size of our interest (5% difference between two conditions) with 95% statistical power and $\alpha$ = 5%. Since we preregistered to exclude incomplete responses and those who failed to correctly answer an attention check question, we recruited 1400 participants from Prolific and had 1397 complete responses ($M_{age}$ = 29.90, SD = 9.73, 702 men, 695 women). Preregistered data exclusion left us with 1393 participants for data analyses. Specifically, we excluded those who did not fully complete the study and those who failed to correctly answer the attention check question, on which they were asked to select *Agree*.

**Procedure.** After giving consent, participants completed two tasks: an artistic preference task (i.e., the minimal group induction) and an economic decision-making task. In the artistic preference task, they were presented with 21 pairs of paintings, one painted by Klee and the other painted by Kandinsky, and were asked to indicate which one they preferred for the 21 pairs. After the task, participants were informed that they belonged either to Group A (Klee) or Group B (Kandinsky) based on their actual artistic preferences. We then asked participants to answer six questions measuring social identification[39], using a scale from 1 = *strongly disagree* to 6 = *strongly agree* (e.g., "Belonging to Group A is an important part of my self-image", $\alpha$ = 0.75).

After the minimal group induction, participants read instructions about the ISDG[35]. In the ISDG, participants were asked to choose between Option X and Option Y, which earned them 3600 and 2700 points, respectively. Participants were further told that there would be other people participating in future studies, and they would complete the same game. Importantly, they were instructed that their decision would influence how much future participants could earn in the game. Specifically, if they choose Option X (i.e., self-payoff maximizing option), a future participant could only earn 2700 and 1800 points by selecting Option X and Option Y, respectively. Contrastingly, if they choose Option Y (sustainable option), a future participant could earn as much as participants could. The ISDG thus represented an intergenerational social dilemma, and the selection of Option Y was our measurement of intergenerational cooperation. Participants answered three comprehension check questions and could not proceed until they correctly answered the questions.

We instructed participants that 10% of participants in Study 1 would receive a bonus payment based on their decision, with the conversion rate of £1 = 2000 points, and, thus, their decision would influence future participants. To avoid deception, we conducted a follow-up study and recruited 140 additional participants (10% of the participants we recruited for the present study). They completed the ISDG as the second generation of the game. If a randomly selected participant from Study 1 made a sustainable choice in the ingroup condition, for instance, we had a participant in the follow-up study from the same minimal group play with the identical payoff structure. Data from this follow-up study was not analyzed.

In the ISDG, we manipulated the group membership of future participants whose decisions would impact. In the ingroup and outgroup conditions, we told participants that future participants would belong to the same and the other artistic preference group (Fig. 1a, b). In the control condition, we did not give any information about the group membership of future participants (Fig. 1c). Participants were randomly assigned to one of the three conditions and then chose between Option X and Option Y, as the first generation of the game.

After making the decision in the ISDG, participants answered questions measuring affinity[3], reputational concern[16,40], legacy motivation[41], and

responsibility[31,32] in a randomized order. For affinity, we modified the four-item scale[3] used by replacing "generations" with "participants", $\alpha$ = 0.78, e.g., "I felt empathetic towards future participants". For legacy motivation, we used two items from the legacy motivation scale[41], $\alpha$ = 0.86, e.g., "It is important to me to leave a positive legacy". For reputational concern, we modified four items from the reputational concern scale[40], $\alpha$ = 0.81, e.g., "During the decision-making task, I thought about how future participants would think about me". For responsibility, we modified and used four items from the previous studies[31,32], $\alpha$ = 0.86, e.g., "I had a responsibility to consider the impact of my decision on future participants." For an exploratory purpose, we included the impact legacy scale[16]. All items were measured with a 7-point Likert scale, from 1 = *not at all* to 7 = *very much so*. Lastly, participants provided demographic information (sex and age).

## Study 2
**Participants, design, and procedure.** The study followed a between-subjects design with three conditions (group: ingroup vs. outgroup vs. control). We preregistered to test the two hypotheses with a logistic regression with the following dummy coding: Dummy 1: ingroup condition vs. control condition (H1); Dummy 2: outgroup condition vs. control condition (H2). Based on the results of Study 1, a priori power analyses revealed that 1707 participants would be sufficient to detect significant planned contrast terms (Dummy 1 and 2). We thus recruited 1750 Japanese participants from Lancers. We had 1805 completed and non-duplicated responses ($M_{age}$ = 42.88, SD = 10.73, 702 men, 695 women). Preregistered data exclusion left us with 1781 participants for data analyses. Specifically, we excluded those who did not fully complete the study and those who failed to correctly answer the attention check question, on which they were asked to select *Agree*.

The study was identical to Study 1 except that the study was written in Japanese, the focal intergroup context was Japan (ingroup) vs. China (outgroup), and the ISDG was not incentivized (i.e., we asked participants to imagine they would complete the ISDG). We note that past research on intergroup cooperation found that the absence (vs. presence) of monetary incentives did not influence results[42]. We chose China as the outgroup as Japan and China share several environmental issues, such as air pollution. In the ISDG, participants were asked to imagine they played the game and instructed that participants in the future generations were Japanese (ingroup), Chinese (outgroup), or those whose nationality was hidden (control). All scales except responsibility ($\alpha$ = 0.58) had satisfactory reliability: affinity: $\alpha$ = 0.74; reputational concern: $\alpha$ = 0.84; legacy motivation: $\alpha$ = 0.87; responsibility: $\alpha$ = 0.74.

## Results
### Study 1
We first looked at the proportion of participants in each experimental condition who chose the sustainable option. In the control condition (N = 463), 226 participants (48.81%) chose the sustainable option. In the ingroup condition (N = 464), 271 participants (58.41%) chose the sustainable option. In the outgroup condition (N = 466), 192 participants (41.20%) chose the sustainable option. Following our preregistration, we first dummy-coded the conditions as follows: Dummy 1: ingroup vs. control; Dummy 2: outgroup vs. control. Using the coding and the decision in the ISDG as a binary dependent variable (unsustainable = 0, sustainable = 1), we conducted a logistic regression. The data met the assumptions. Supporting H1, we found a significant effect of Dummy 1, B = 0.39, 95% CI [0.12, 0.65], p < 0.001 (all analyses were two-tailed tested). Moreover, supporting H2, we found a significant effect of Dummy 2, B = -0.31, 95% CI [−0.57, −0.05], p = 0.02. Thus, we found both ingroup favoritism and outgroup derogation. The results held after controlling for the effect of age and sex. We note that the mean social identification score was 3.31 (SD = 0.09), and it was below the scale midpoint. While the induced social identification was not particularly strong, we found the effect of group membership on sustainable decision-making. This echoes the past scholarship, consistently demonstrating that the strength of social identification does not

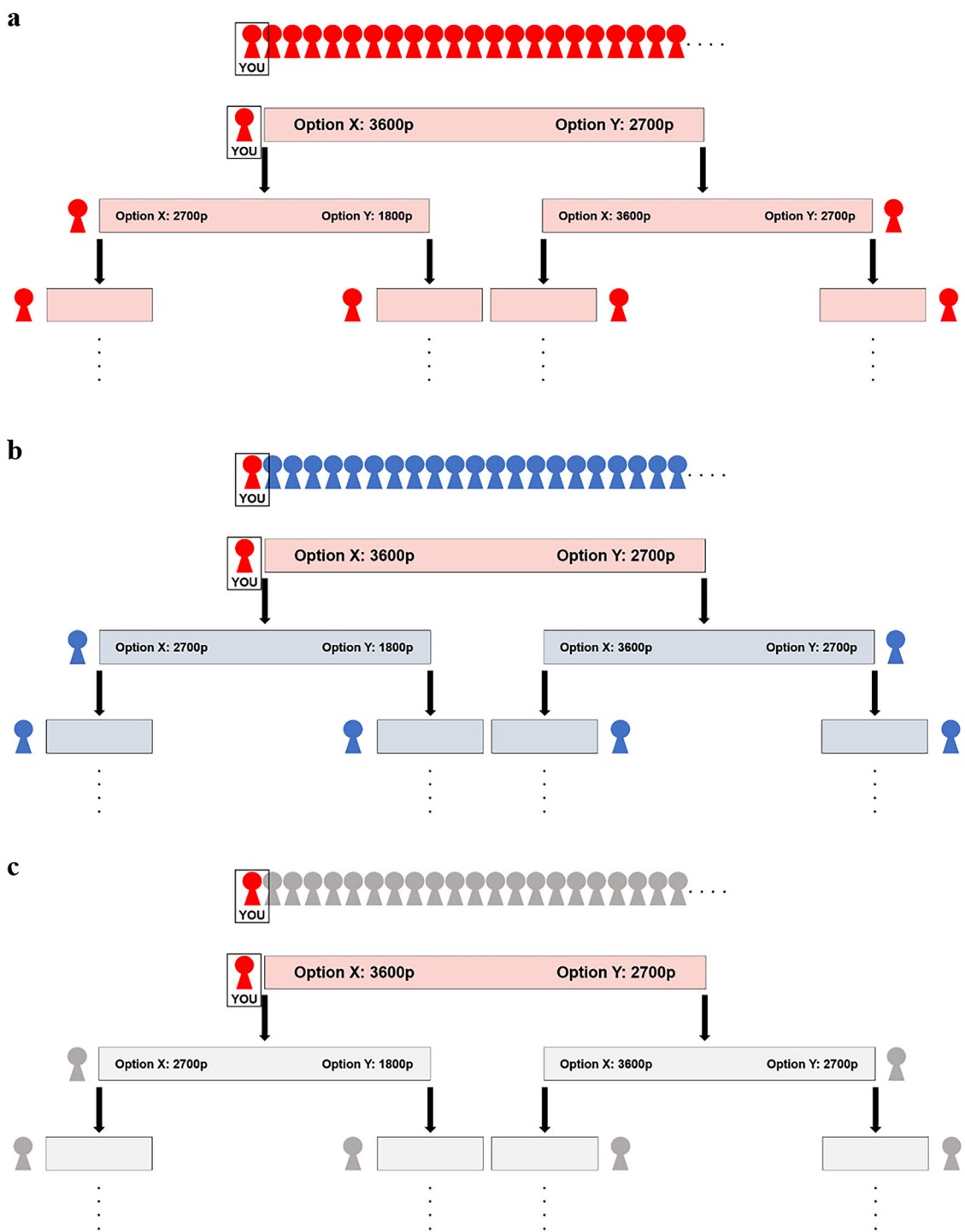

**Fig. 1 | Graphical Summary of Experimental Conditions. a** Ingroup condition. **b** Outgroup condition. **c** Control condition.

translate into ingroup favoritism in non-intergenerational social dilemmas[5,6].

Following our preregistration, we examined the effect of group on affinity, reputational concern, legacy motivation, and responsibility (see Table 1 for descriptive statistics), using the dummy coding. The normality assumption was violated for the four dependent variables and the equal variance assumption was violated for affinity, reputational concern, and responsibility. We thus report the results of robust regression analyses in the online supplementary results. The results of the preregistered regression analyses and the robust regression analyses did not differ in any meaningful ways. Table 2 summarizes the results of the four regression analyses. Data

met the assumptions except the normality assumption, and we report the results of robust regression analyses in Supplemental Material (S1). The results of the preregistered regression analyses and the robust regression analyses did not differ in any meaningful ways. Overall, we found that individuals scored higher on affinity, legacy motivation, and responsibility when the future generations belonged to their ingroup compared to when they did not know about the group membership of the future generations. Contrastingly, we found a significant difference in reputational concern between the outgroup and control conditions, suggesting that the increased selfishness in the outgroup condition might be explained by the lowered reputational concern they felt.

Finally, we examined the psychological processes underlying the observed ingroup favoritism and outgroup derogation. Specifically, we ran a mediation analysis where Dummy 1 predicted the sustainable decision-making via affinity, legacy motivation, and responsibility, the variables that were significantly predicted by Dummy 1. We bootstrap-tested the indirect effects (bootstrap = 1000) and found that the indirect effect via responsibility was significant, $B = 0.09$, 95% CI [0.04, 0.13], $p < 0.001$. The other mediation effects were not significant: affinity: $B = 0.007$, 95% CI [0.0004, 0.02], $p = 0.12$; legacy: $B = -0.002$, 95% CI [−0.008, 0.002], $p = 0.40$. The direct effect was not significant, $B = 0.01$, 95% CI [−0.05, 0.05], $p = 0.84$. Thus, the effect of shared group membership with future generations was fully mediated by increased responsibility for future generations. We further carried out a causal mediation analysis[43] (the number of simulations = 1000, quasi-Bayesian Monte Carlo Method), and found a significant average causal mediation effect of responsibility, controlling for the effects of affinity and legacy, $b = 0.59$ [0.32, 1.01], $p < 0.001$.

Next, we tested the mediation model where Dummy 2 predicted the sustainable decision-making via reputational concern. The indirect effect was significant, $B = -0.06$, 95% CI [−0.04, −0.003], $p = 0.03$. The direct effect was not significant ($B = -0.02$, 95% CI [−0.12, 0.004], $p = 0.06$), and the effect of outgroup membership was fully mediated by the lowered level of reputational concern that individuals felt. The causal mediation analysis also revealed the significant average mediation effect of reputational concern, $b = 0.23$ [0.03, 085], $p = 0.03$.

Overall, we did find support for our hypotheses, revealing ingroup favoritism and outgroup derogation in the intergenerational context. Our findings thus extended the previous literature on intergroup cooperation to the intergenerational context. It is noteworthy that we observed outgroup derogation, where individuals are less likely to choose the sustainable option when the future generation belongs to the outgroup. Furthermore, Study 1 provides insights into the psychological underpinnings of intergenerational intergroup cooperation. Specifically, the mediation analyses suggest that it is responsibility and reputational concern that account for the effect of group membership in intergenerational intergroup cooperation; individuals are more likely to make a sustainable decision for future generations that share the group membership as they feel an increased level of responsibility for them. Contrastingly, they are less likely to make a sustainable decision when future generations that their action influences belong to an outgroup because they feel a reduced level of reputational concern when making the decision.

## Study 2

In the control condition ($N = 590$), 250 participants (42.37%) chose the sustainable option. In the ingroup condition ($N = 600$), 326 participants (54.33%) chose the sustainable option. In the outgroup condition ($N = 591$), 158 participants (26.73%) chose the sustainable option. We conducted a logistic regression analysis. The data met the assumptions. Supporting H1, we found a significant effect of Dummy 1, $B = 0.48$, 95% CI [0.25, 0.71], $p < 0.001$. Supporting H2, we found a significant effect of Dummy 2, $B = -0.70$, 95% CI [−0.95, −0.46], $p < 0.001$. Replicating Study 1, we found both ingroup favoritism and outgroup derogation. We note that the effect size of outgroup derogation was substantially bigger than that in Study 1, the minimal group context. The results held after controlling for the effect of age and sex.

Table 3 summarizes descriptive statistics of the key psychological variables by condition. The normality and homoscedasticity assumptions, and we report the results of robust regression analyses in the online supplementary results (see S2). The results of the preregistered regression analyses and the robust regression analyses did not differ in any meaningful ways. We found that Dummy 2 significantly influenced affinity, reputational concern, legacy motivation, and responsibility (Table 4) such that outgroup membership reduces the extent to which participants felt affinity, reputational concern, legacy motivation, and responsibility. Contrastingly, Dummy 1 (i.e., shared ingroup membership with future generations) significantly influenced only responsibility.

To elucidate the psychological processes underlying the observed ingroup favoritism and outgroup derogation, we conducted mediation analyses. We first built a model in which Dummy 1 predicted the sustainable decision via responsibility. Following Study 1, we bootstrapped-tested the indirect effect. Consistent with Study 1, we found that responsibility significantly mediated the relationship between shared group membership and sustainable decision-making, $B = 0.07$, 95% CI [0.04, 0.11], $p < 0.001$. The direct effect was also significant, $B = 0.05$, 95% CI [0.01, 0.09], $p = 0.04$. The causal mediation analysis also revealed the significant average mediation effect, $b = 0.61$ [0.38, 1.00], $p < 0.001$.

Regarding outgroup derogation, we found that affinity ($B = -0.01$, 95% CI [−0.03, −0.01], $p = 0.009$), legacy motivation ($B = -0.01$, 95% CI [−0.02, −0.002], $p = 0.02$), and responsibility ($B = -0.03$, 95% CI [−0.06, −0.01], $p = 0.01$), but not reputational concern ($B = 0.002$, 95% CI [−0.002, 0.01], $p = 0.40$), significantly mediated the relationship between outgroup membership and sustainable decision-making. These mediation effects were replicated by causal mediation analyses: affinity: $b = 0.10$ [0.04, 0.24], $p = 0.046$; legacy: $b = 0.07$ [0.01, 0.16], $p = 0.01$; responsibility: $b = 0.27$ [0.07, 0.46], $p = 0.01$. Outgroup membership was negatively associated with felt affinity, legacy motivation, and responsibility, in turn reducing intergenerational cooperation. The direct effect was significant, $B = -0.17$, 95% CI [−0.31, −0.05], $p = 0.01$.

Overall, we found future ingroup favoritism and outgroup derogation in the actual group context. It thus suggests that ingroup favoritism and outgroup derogation in intergenerational sustainable decision-making are robust phenomena, and the negative impact of outgroup membership can be exacerbated in actual intergroup contexts. Regarding the psychological mechanisms, consistent with Study 1, we found that responsibility played a critical role in bridging the relationship between shared ingroup membership with future generations and sustainable decision-making. In the

**Table 1 | Descriptive statistics of affinity, reputational concern, legacy motivation, and responsibility by condition (Study 1)**

|  | Ingroup | Outgroup | Control |
|---|---|---|---|
| Affinity | 4.94 (1.25) | 4.70 (1.31) | 4.69 (1.31) |
| Reputational concern | 3.36 (1.53) | 3.07 (1.43) | 3.31 (1.50) |
| Legacy motivation | 5.34 (1.44) | 5.06 (1.52) | 5.09 (1.49) |
| Responsibility | 4.50 (1.52) | 4.08 (1.56) | 4.08 (1.57) |

Note: numbers in brackets indicate standard deviations.

**Table 2 | Results of regression analyses on affinity, reputational concern, legacy motivation, and responsibility (Study 1)**

|  | F statistics | Dummy 1 | Dummy 2 |
|---|---|---|---|
| Affinity | $F_{(2, 1390)} = 5.45$, $p < 0.001$ | 0.25* [0.08, 0.41] | 0.01 [−0.15, 0.18] |
| Reputational concern | $F_{(2, 1390)} = 4.95$, $p = 0.01$ | 0.05 [−0.14, 0.25] | −0.23* [−0.43, −0.04] |
| Legacy motivation | $F_{(2, 1390)} = 4.95$, $p = 0.01$ | 0.25* [0.05, 0.44] | −0.03 [−0.23, 0.16] |
| Responsibility | $F_{(2, 1390)} = 11.41$, $p < 0.001$ | 0.42* [0.22, 0.62] | 0.002 [−0.20, 0.20] |

Note: numbers in square brackets indicate lower and upper bounds of the 95% confidence interval. *$p < 0.05$. Dummy 1: ingroup (1) vs. control (0). Dummy 2: outgroup (1) vs. control (0).

present study, we found that affinity, legacy motivation, and responsibility accounted for outgroup derogation in the Japan vs. China context, inconsistently with Study 1. This suggests that the psychological underpinnings of future outgroup derogation in natural intergroup contexts can be complex and context-dependent.

## Discussion

In the present research, we investigated whether shared group membership between current and future generations would foster intergenerational cooperation (i.e., sustainable behavior) using the minimal and national intergroup context and the ISDG. We hypothesized that people would be more and less cooperative with future ingroup members and outgroup members, respectively. Supporting the hypotheses, we found ingroup favoritism and outgroup derogation in both minimal and national intergroup contexts; shared ingroup membership and outgroup membership promote and lower intergenerational cooperation, respectively. In addition, we revealed that increased responsibility consistently mediated the effect of shared group membership on intergenerational cooperation in both the minimal and nationality contexts. Yet, the psychological processes associated with the effect of outgroup membership on intergenerational cooperation were somewhat different in the minimal and nationality contexts.

Our finding extends the extensive literature on ingroup favoritism[4,44] by demonstrating it in the intergenerational contexts and by revealing contributors to future ingroup favoritism, perceived responsibility. The result suggests that ingroup framing can be an effective strategy to promote intergenerational cooperation. Previous studies on proenvironmental behavior suggested that proenvironmental identity plays a pivotal role in shaping proenvironmental behavior[45–47]. Fostering a new identity, especially when it conflicts with existing ones, can be challenging. Highlighting the consequences of present behavior on future ingroup members does not require identity change and can be an efficient way to promote sustainable behavior.

We consistently found outgroup derogation in the intergenerational dilemma, while outgroup derogation is rarely observed in non-intergenerational social dilemmas[4,48,49]. In non-intergenerational dilemmas, previous studies have robustly shown that people prefer not doing anything over harming outgroup members[50] and they thus do not discriminate between outgroup members and strangers when making cooperative decisions. We found that in the minimal intergenerational context, individuals experience a lower level of reputational concern, and this was then associated with a reduced willingness to be sustainable. This finding contributes to the previous literature on reputation and sustainable behavior, which has collated mixed evidence[51,52]. Our result suggests that

the role of reputational concern in shaping sustainable behavior is conditional on whom one's sustainable behavior is believed to influence. Contrastingly, however, in the natural group context, we did not find evidence that reputational concern significantly mediated the relationship between outgroup membership and intergenerational cooperation. Thus, we suggest that the psychological mechanisms underlying the unwillingness to make sustainable decisions for future outgroup members may be context-dependent when it comes to natural group contexts.

Nevertheless, we conjecture that there are alternative explanations for the discrepancy between non-intergenerational and intergenerational contexts. One possible explanation is that the temporal distance between participants and future outgroup members might reduce the sense of moral harm caused by outgroup derogation. The second possibility is that people's sense of power (i.e., being able to unilaterally influence future generations) uniquely decreased cooperation with future outgroup members. Previous studies suggested that having complete power over others induces a sense of responsibility and promotes generosity[53] in general. On the other hand, having power leads to outgroup dehumanization and, correspondingly, justifies harm against outgroup members[54,55]. Thus, in intergenerational social dilemmas, power over future outgroup members may reduce intergenerational outgroup cooperation via increased dehumanizing tendencies towards outgroup members. We argue that psychological processes underlying outgroup derogation in intergenerational social dilemmas deserve further research.

Our findings offer important practical implications for existing intervention approaches that highlight the immediate impacts of climate change on specific regions of the world. While it is important to acknowledge that climate change impartially impacts different countries[56], highlighting those countries and encouraging individuals to act for them can rather be counterproductive. For instance, climate change causes sea level rise, and the flood and land subsidence risks are particularly high for some cities and areas. As such, there have been campaigns and media coverage highlighting such high-risk areas to increase the awareness of climate change (e.g., Tuvalu). Our finding suggests that these attempts may backfire; when people believe that their immediate sustainable behavior mainly benefits the future of an outgroup and its members, they may be less likely to engage in sustainable behavior. In sum, despite that climate change is indeed a global issue, it may be more effective to frame it as a local issue, encouraging individuals to think about the future of their own group. In our research, we did not compare the effect of ingroup and outgroup membership labeling against that of universal labeling (i.e., participants know their decision is going to influence everyone in the future, regardless of their group membership). Previous psychological work suggests that the activation of a common superordinate identity improves intergroup relations[57,58] and such a universal labeling may promote intergenerational cooperation that benefits outgroup members without backlash.

In addition, our findings are relevant to instances where one country's selfish actions harm the future of other nations, without facing any immediate consequences itself. Historically, developed countries and their industries have implemented a range of policies and practices that substantially harm the ecosystem of other nations (e.g., dam projects disrupting water flow to downstream nations and their local ecosystem, exporting of electronic wastes to other countries, exploitation of critical minerals from developing countries while causing local environmental destruction, etc.). Consistent outgroup derogation observed in our studies speaks to the

**Table 3 | Descriptive statistics of affinity, reputational concern, legacy motivation, and responsibility by condition (Study 2)**

|  | In-group | Out-group | Control |
|---|---|---|---|
| Affinity | 4.04 (1.17) | 3.63 (0.99) | 3.94 (1.16) |
| Reputational concern | 3.39 (1.33) | 3.01 (1.24) | 3.25 (1.41) |
| Legacy motivation | 4.14 (1.41) | 3.75 (1.42) | 4.01 (1.51) |
| Responsibility | 3.74 (1.27) | 3.27 (1.12) | 3.44 (1.22) |

Note: numbers in brackets indicate standard deviations.

**Table 4 | Results of regression analyses on affinity, reputational concern, legacy motivation, and responsibility (Study 2)**

|  | F statistics | Dummy 1 | Dummy 2 |
|---|---|---|---|
| Affinity | $F_{(2, 1778)} = 21.68$, $p < 0.001$ | 0.10 [−0.02, 0.23] | −0.30* [−0.43, −0.18] |
| Reputational concern | $F_{(2, 1778)} = 12.39$, $p < 0.001$ | 0.14 [−0.01, 0.29] | −0.24* [−0.39, −0.08] |
| Legacy motivation | $F_{(2, 1778)} = 11.44$, $p < 0.001$ | 0.13 [−0.03, 0.30] | −0.26* [−0.43, −0.10] |
| Responsibility | $F_{(2, 1778)} = 23.54$, $p < 0.001$ | 0.30* [0.16, 0.44] | −0.17* [−0.31, −0.04] |

Note: numbers in square brackets indicate lower and upper bounds of the 95% confidence interval. * $p < 0.05$. Dummy 1: ingroup (1) vs. control (0). Dummy 2: outgroup (1) vs. control (0).

prevalence of such cases, and our findings on psychological underpinnings may help us design effective interventions. Psychological messaging has been identified as an effective tool to change behavior[59], and therefore, communications highlighting and triggering legacy motivations and social responsibility for the future may promote intergenerational cooperation for the future of outgroups.

## Limitations

Lastly, we would like to discuss the limitations of the research and future directions. While the ISDG offers a clear measurement of sustainable vs. unsustainable self-serving choices, the intergenerational social dilemma that it represents does not fit all forms of actual intergenerational dilemmas we face. In the ISDG, for instance, it is an individual decision rather than a collective decision that impacts future generations. Thus, our findings may not be generalized to explain decision-making in intergenerational dilemmas in which individuals' decision-making cumulatively impacts future generations[60]. While economic games abstract and stylize complex social environments, they are indeed different from actual pro-environmental and sustainable decision-making. As such, it is sensible to examine whether our findings are generalizable to a diverse set of actual sustainable decisions.

In our work, we did not consider individual difference factors. On the one hand, our findings suggest the robust effect of group membership on intergenerational cooperation, implying broad policy applicability regardless of individual differences. On the other hand, given the prevalence and effectiveness of microtargeting in political communication[61,62], future research could explore individual differences closely to determine whether and how much group framing influences intergenerational cooperation across individuals.

## Data availability

All data can be found at https://osf.io/zt3k4/.

## Code availability

Analyses codes can be found at https://osf.io/zt3k4/.

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

## Acknowledgements

This work is supported by Founders Pledge Incorporated, but the funder had no role in study design, data collection and analysis, decision to publish or preparation of the paper.

## Author contributions

H.I.: conceptualization, methodology, validation, investigation, resources, data curation, writing—original draft, project administration; Y.I.: conceptualization, methodology, validation, writing—review & editing; A.Y.W.: methodology, resources, writing—review & editing; T.S.: writing—review & editing; N.M.: conceptualization, methodology, writing—review & editing.

## Competing interests

The authors declare no competing interests.
