## [Transparent Peer Review file · Communications Psychology]

Intergenerational Intergroup Cooperation: Future ingroup favoritism and outgroup derogation in the minimal and natural group contexts

Corresponding Author: Dr Hiroataka Imada

Version 0:

Decision Letter:

Dear Dr Imada,

Thank you for your patience during the peer-review process. Your manuscript titled "Intergenerational Intergroup Cooperation: "Future" ingroup favoritism and outgroup derogation in the minimal and natural group contexts" has now been seen by 2 reviewers, and I include their comments at the end of this message. They find your work of interest but raised some important points. We are interested in the possibility of publishing your study in Communications Psychology, but would like to consider your responses to these concerns and assess a revised manuscript before we make a final decision on publication.

We therefore invite you to revise and resubmit your manuscript, along with a point-by-point response to the reviewers. Please highlight all changes in the manuscript text file.

Editorially, we encourage you to provide additional details and discuss the limitations of the study concerning the use of deception and external validity, as highlighted by Reviewer 2. Additionally, please improve your statistical reporting to align with the journal's guidelines and Reviewer 1's recommendations.

I am attaching an Editorial Requests Table that details critical reporting requirements for the revised manuscript. Please attend to each item and ensure your manuscript is fully compliant. If your revised manuscript is not aligned with these requests on major issues, such as those concerning statistics, it may be returned to you for further revisions without re-review.

Please submit the following items:

- Revised manuscript
- Point-by-point response to the referees' comments
- Cover letter (as a separate document)
- <https://www.nature.com/documents/nr-reporting-summary.zip>>Nature Research Reporting Summary
- <https://www.nature.com/documents/nr-editorial-policy-checklist.pdf>>Editorial Policy Checklist
- Completed Editorial Request Table (attached).

via this link: Link Redacted .

Additional guidance is available in our style and formatting guide Communications Psychology formatting guide.

Best regards,

Yafeng Pan

Yafeng Pan, PhD
Editorial Board Member
Communications Psychology
orcid.org/0000-0002-5633-8313

REVIEWER EXPERTISE:

Reviewer #1: intergenerational cooperation, ingroup/outgroup interactions, large-scale survey

Reviewer #2: ingroup/outgroup interactions, large-scale survey

REVIEWER REPORTS:

Reviewer #1 (Remarks to the Author):

Please see the attached review. I think this manuscript has a lot of promise. See my comments for thoughts on how certain potential issues could be addressed and how the addition of specific points of information could be of use.

Title

I do not see a reason why the word "future" should be in quotations. It's meaning does not seem different because of it, so I would remove the quotations from the title.

Abstract

Starting at line 6, the abstract becomes hard to understand. That sentence is grammatically incorrect, then on the following sentence on line 9 the combination of more and less likely in the same sentence is confusing. Please split these into separate sentences.

Introduction

1. I like the framing used in the first paragraph of the Introduction, but I also want to highlight that climate change and other issues are not entirely zero-sum. There are certainly instances where an action taken for the future could also be beneficial for the present. For instance, preserving a forest and protecting animals (or natural beauty in general). I think it's worth acknowledging this caveat to the author(s) overarching and broad-sweeps claims about the issue.

2. For the paragraph starting at line 45, it is also worth acknowledging that it is possible for someone to be acknowledged posthumously. In fact, legacy motivation is a strong driver of sustainable behaviors (see work by Markowitz and colleagues, Syropoulos and colleagues). So I think that although reputation might not be the primary driver, wanting to be remembered as a good person by future people could be, for some, a motivator.

3. I think line 63 is not an exhaustive listing of the relevant variables predicting intergenerational decision-making. Some candidate constructs that come to mind include: impartial intergenerational beneficence, legacy motivations/concerns, future self-continuity, consideration of future consequences, and moral obligation/responsibility to future generations. A review of these can be found in this recent paper:

Law, K. F., Colaizzi, G., & Syropoulos, S.(2025). Climate change is an intergenerational challenge that requires

intergenerationally focused behavioral solutions. *Current Opinion in Behavioral Sciences*, 71, 101467.
<https://doi.org/10.1016/j.cobeha.2024.101467>

4. I don't agree with the authors that legacy motives and responsibility fall under the umbrella of generativity. I have not seen that argument before. It is an interesting one, but ultimately, my reading of the literature is that generativity is a construct that is more biologically rooted, and one that focuses on one's own kin and the life stage during which people start feeling a motivation to help improve the lives of their own descendants. Legacy is more broadly concerned about how one's remembered. A person without kids can have legacy concerns, and concerns that are motivated by prosocial or egotistical motives. Similarly, responsibility to future generations is also a sense of obligation felt to protect future people more broadly, and not just one's own descendants. I think these should be distinguished from the umbrella of generativity and discussed as separate and related constructs. See an example in Syropoulos and colleagues. Syropoulos, S., Mah, A., & Markowitz, E. (2023). Personal legacy: A psychological mechanism for increasing climate action and coping with climate change stressors. *Journal of Mental Health and Climate Change*, 1, 16-29.
<http://doi.org/10.5281/zenodo.8320457>

5. Line 85. Please provide a citation for the definition of legacy.

Study 1

1. I really liked the design of Study 1, and felt that the authors provided a very clear explanation of it. I think Table 1 is a bit confusing. I am not sure what the third column means. So I would recommend revising it for clarity, or simply removing that table as the Figures that follow capture the same information.

2. I think for Table 3, it'd be very useful to add effect sizes (maybe Cohen's *d*) for each comparison. Also, adding ** for $p < .01$, and *** for $p < .001$ would be a nice addition too.

3. I appreciate the mediation test, but I have a question here. What were the correlations between all variables? It would be good to know, in case there are multi-collinearity issues.

Study 2

I have the same comments regarding effect sizes, correlations between variables, and *p* values. Other than these, the design was a nice addition and replication of Study 1 in an intergroup context.

Discussion

I enjoyed the discussion. I think the only addition here would be the authors' speculation of what we can do to overcome this outgroup derogation and ingroup favoritism effect in intergenerational decisions. This could spark interesting directions in the field of study.

Minor Comments

1. Rather than saying "more specifically" simply say "specifically". The word more does not add any additional value.

Reviewer #2 (Remarks to the Author):

REVIEW OF COMMSPSYCHOL-25-0028
["Intergenerational Intergroup Cooperation"]

SUMMARY: The paper presents two studies in which participants play 'intergenerational' economic games with either ingroup, outgroup, or neutral recipients in later generations. The paper's main finding is that, participants exhibit ingroup favoritism and outgroup derogation also when playing in this intergenerational setting.

EVALUATION: To the best of my knowledge the research design is novel and represents an interesting addition to the literature concerning ingroup favoritism; moreover, the findings appear replicable and robust to group context (minimal vs. national). The main weaknesses of the paper are an unnecessary use of deception and a lack of external validity.

RECOMMENDATION: reject

MAIN REMARKS:

(A) Section 8.07a of the APA's Ethics Code reads: "Psychologists do not conduct a study involving deception unless they have determined that the use of deceptive techniques is justified by the study's significant prospective scientific, educational, or applied value and that effective nondeceptive alternative procedures are not feasible." The present study violates this code in at least four ways: (i) participants are led to believe that multiple (!) 'generations' of other participants will be involved in the study (see page 39/51 of the experimental instructions), which is not the case. There is only one future generation. It would have been easily feasible to include a third generation in Studies 1 and 2, i.e., to not deceive participants. (ii) Additionally, for Study 2 it is not clear to me if the authors invited participants for the second generation at all. (iii) It remains further unclear if the second generation in both studies were paid and if (iv) they were actually composed of ingroup or outgroup members.

All four mentioned problems indicate unnecessary deception towards the participants. They are severe enough to make me recommend rejection of this paper. I'm aware, though, that this is a strict reading of the Ethics Code and so I'm fine with

having my opinion be overruled by a less strict view of the editor and/or a majority of other reviewers.

(B) External validity: I doubt that using an ISDG in an online setting adequately approximates intergenerational dynamics. Its sequence resembles that of a standard dictator game and thus the results might not be interpretable as intergenerational at all: what is missing completely in the design is the present generation's stake in the outcomes of the future generation (usually: parents would care for their kids' wellbeing). A refined experimental design would add such a dimension (ideally as an experimental condition) and would further test whether the 'generation'-story matters at all by removing the deceptive third and later generations to test for any effect of such time horizon. Finally: what type of situation do the authors have in mind with their outgroup condition? Which realistically relevant actions can I take which only affect outgroup members in the future? Climate change, for example, affects both in- and outgroup future generations. Same for most (all?) other environmental dilemmas.

Thus, it remains unclear if the choice of ISDG allows for the results to be generalized to the broad societal implications the paper lists in both introduction and discussion. This is also admitted by the authors themselves. Accordingly, the authors might want to discuss more fitting societal applications for their experimental design.

Statistics:

The chosen statistical models are to the best of my knowledge applicable for the intended interpretation and the research objectives. The hypotheses are preregistered, and the interpretation of the data is well done. However, given that choosing the cooperative option in every generation would maximize social welfare, efficiency and social welfare concerns of the participants could have been taken into account when eliciting and analyzing the data.

MINOR COMMENTS:

The authors should provide additional data to show that the MGI-task induced social identification in the participants. (lines 169 - 172)

The procedure contains a mistake: 10% of the 1400 participants of Study 1 are 140, not as stated 60. If these 60 participants were selected from the pilot data this should be clearly stated. (Lines 188 - 189).

There are many spelling and grammatical mistakes throughout, for example in lines 161, 166, 167 and 367.

If you experience problems in linking your ORCID, please contact the Platform Support Helpdesk.

Version 1:

Decision Letter:

Dear Dr Imada,

Your manuscript titled "Intergenerational Intergroup Cooperation: "Future" ingroup favoritism and outgroup derogation in the minimal and natural group contexts" has now been seen by our reviewer, whose comments appear below. In light of their advice I am delighted to say that we are happy, in principle, to publish a suitably revised version in Communications

Psychology.

We therefore invite you to revise your paper one last time to address the remaining concerns of our reviewer and a list of editorial requests. At the same time we ask that you edit your manuscript to comply with our format requirements and to maximise the accessibility and therefore the impact of your work.

EDITORIAL REQUESTS:

SUBMISSION INFORMATION:

OPEN ACCESS:

* DATA AVAILABILITY:

Link Redacted

Best regards,

Jennifer Bellingtier

Jennifer Bellingtier, PhD
Senior Editor
Communications Psychology

Yafeng Pan, PhD
Editorial Board Member

REVIEWER EXPERTISE:

Reviewer #2: ingroup/outgroup interactions, large-scale survey

REVIEWERS' COMMENTS:

Reviewer #2 (Remarks to the Author):

REVIEW OF COMMSPSYCHOL-25-0028A
["Intergenerational Intergroup Cooperation"]

SUMMARY: The paper presents two studies in which participants play 'intergenerational' economic games with either ingroup, outgroup, or neutral recipients in later generations. The paper's main finding is that participants exhibit ingroup favoritism and outgroup derogation also when playing in this intergenerational setting.

EVALUATION: To the best of my knowledge the research design is novel and represents an interesting addition to the literature concerning ingroup favoritism; moreover, the findings appear replicable and robust to group context (minimal vs. national). The revision addressed most of my concerns and I recommend to accept this contribution after very minor revisions (see below).

RECOMMENDATION: minor revision

MAIN REMARKS AFTER REVISION:

Deception:

The revised text improved the clarity of the experimental method. While I still view the instructions and graphics shown to the participants as somewhat deceptive regarding the existence of multiple future generations, in its current form the study's strengths and weaknesses are clear and can thus be assessed by each reader individually. The clarification that Study 2 is an unincentivized vignette is sufficient, as is the added clarification on the payment of the second generation in Study 1. I would however further add a description of how the second generation in Study 1 was matched. This was well formulated in the authors' response to my initial review but is still absent from the paper's text.

External Validity:

The provided clarification shows that ISDGs are commonly applied in these settings.

Discussion:

The added examples in the discussion point to fitting real-world equivalents of the stylized decisions of the ISDG.

Welfare Effects:

The revision makes it clear that the mentioned individual welfare considerations were not of interest for the authors' research objectives, the provided clarification suffices. I would have considered a differentiation of these effects as a further improvement of the contribution, but this can also be achieved as part of further examinations in future projects.

Reviewer #1 (Remarks to the Author):

Please see the attached review. I think this manuscript has a lot of promise. See my comments for thoughts on how certain potential issues could be addressed and how the addition of specific points of information could be of use.

[Response] Thank you very much for taking your time to review our manuscript and for the constructive feedback. We have carefully consumed and addressed your comments below:

Title

I do not see a reason why the word “future” should be in quotations. It’s meaning does not seem different because of it, so I would remove the quotations from the title.

[Response] In previous studies on non-intergenerational intergroup cooperation, outgroup derogation was consistently absent. Nevertheless, we observed strong and consistent outgroup derogation in the intergenerational contexts. Thus, with the quotations, we intended to highlight that we observed both ingroup favoritism and outgroup derogation in the intergenerational, *future* context. That being said, we do not have a strong preference and we are happy to drop it. Since it is a stylistic issue, we would like to leave the decision for the editor.

[Revision] No revisions are made (yet).

Abstract

Starting at line 6, the abstract becomes hard to understand. That sentence is grammatically incorrect, then on the following sentence on line 9 the combination of more and less likely in the same sentence is confusing. Please split these into separate sentences.

[Response] Thank you very much for your careful reading.

[Revision] We revised the abstract and it reads as follows (revised parts are italicized):

“Issues related to sustainability (e.g., climate change and over-fishing) manifest themselves as intergenerational social dilemmas, and people are constantly faced with a choice between self-serving unsustainable behavior and sustainable, personally costly behavior. Extending the previous literature on (non-intergenerational) intergroup cooperation, we tested whether group membership of the future generations influenced sustainable decision making. *In two preregistered studies using the intergenerational sustainability dilemma game, we found that individuals were more and less likely to make a sustainable (vs. selfish) decision when they believed that their current behavior would benefit future ingroup and outgroup members, respectively, in the minimal group (Study 1: N = 1393) and the national group (Study 2: Japan vs. China, N = 1781) contexts.* Future ingroup favoritism and outgroup derogation were primarily driven by the increased felt responsibility for future generations and the reduced sense of reputational concern.”

Introduction

1. I like the framing used in the first paragraph of the Introduction, but I also want to highlight that climate change and other issues are not entirely zero-sum. There are certainly instances where an action taken for the future could also be beneficial for the present. For instance, preserving a forest and protecting animals (or natural beauty in general). I think it’s

worth acknowledging this caveat to the author(s) overarching and broad-sweeps claims about the issue.

[Response] Thank you for your thoughts. Indeed, some behaviors and policies benefit both the current and future generations. When the interests of the two are aligned, there are not conflict of interests and such situations are no longer characterized as social dilemmas. As such, behaviors that benefit both are not intergenerational cooperation and are beyond the scope of our research. Previous studies have found that messaging and communications about self-benefits of pro-environmental actions successfully promote proenvironmental behaviors (Van de Vyver et al., 2018) and such behaviors are easier to promote than intergenerational cooperation in intergenerational social dilemmas, justifying why we studied the latter rather than the former.

This being said, we admit that our introduction reads as if all sorts of sustainability issues fall under social dilemma that we study and we believe clarification is needed.

[Revision] In the first paragraph, we made a minor revision (revised parts are italicized): “Many societal issues manifest themselves as social dilemmas, situations in which personal and collective interests conflict with each other^{1,2}. *Some sustainability issues have such an underlying structure, and individuals are often faced with a choice between self-benefit maximizing unsustainable behavior and sustainable behavior that benefits future generations (i.e., intergenerational social dilemmas³).* Some issues are global (e.g., climate change), but others are local (e.g., overfishing), which makes the marked psychological difference; in local intergenerational social dilemmas, sustainable behavior is targeted only at future generations of one’s own local community.”

2. For the paragraph starting at line 45, it is also worth acknowledging that it is possible for someone to be acknowledged posthumously. In fact, legacy motivation is a strong driver of sustainable behaviors (see work by Markowitz and colleagues, Syropoulos and colleagues). So I think that although reputation might not be the primary driver, wanting to be remembered as a good person by future people could be, for some, a motivator.

[Response] As we indeed measured reputation legacy, we agree with you that individuals do care about how they are remembered and seen by future generations (whether or not they themselves are alive). Yet, this form of reputational concern is distinct from that discussed in the previous literature on indirect reciprocity and non-intergenerational cooperation in that individuals do not materialistically or financially benefit from being seen as a nice person by individuals in the future generations. We have read our writing again and we believe that it is clear that we are not discussing reputation legacy but arguing that reputational concern as assumed in indirect reciprocity is absent in intergenerational social dilemmas.

Readers with prior knowledge about past work on intergenerational cooperation may have the same thought at line 45, but they will find our discussions about reputation legacy and a couple of paragraphs later and our results associated with reputational concern.

[Revision] No revisions were made.

3. I think line 63 is not an exhaustive listing of the relevant variables predicting intergenerational decision-making. Some candidate constructs that come to mind include: impartial intergenerational beneficence, legacy motivations/concerns, future self-continuity, consideration of future consequences, and moral obligation/responsibility to future

generations. A review of these can be found in this recent paper:

Law, K. F., Colaizzi, G., & Syropoulos, S.(2025). Climate change is an intergenerational challenge that requires intergenerationally focused behavioral solutions. *Current Opinion in Behavioral Sciences*, 71, 101467. <https://doi.org/10.1016/j.cobeha.2024.101467>

[Response] Thank you very much for those ideas. We believe that we have in fact covered legacy, consideration of future consequences and moral obligation/responsibility in our study (as legacy, generativity, and responsibility respectively).

Regarding impartial intergenerational beneficence and future self-continuity, these are individual difference factors (e.g., Syropoulos et al., 2025) that we did not focus on in our research. Our thinking behind this decision was that when thinking about policy impact, individual differences are almost always not considered (except for cases of microtargeting: e.g., Tappin et al., 2023).

[Revision] We received a similar comment from Reviewer 2 regarding the potential role of individual differences and we added the following in the end of the manuscript: “In our work, we did not consider individual difference factors. On the one hand, our findings suggest the robust effect of group membership on intergenerational cooperation, implying broad policy applicability regardless of individual differences. On the other hand, given the prevalence and effectiveness of microtargeting in political communication^{60,61}, future research could explore individual differences closely, to determine whether and how much group framing influences intergenerational cooperation across individuals.”

4. I don't agree with the authors that legacy motives and responsibility fall under the umbrella of generativity. I have not seen that argument before. It is an interesting one, but ultimately, my reading of the literature is that generativity is a construct that is more biologically rooted, and one that focuses on one's own kin and the life stage during which people start feeling a motivation to help improve the lives of their own descendants. Legacy is more broadly concerns about how one's remembered. A person without kids can have legacy concerns, and concerns that are motivated by prosocial or egotistical motives. Similarly, responsibility to future generations is also a sense of obligation felt to protect future people more broadly, and not just one's own descendants. I think these should be distinguished from the umbrella of generativity and discussed as separate and related constructs. See an example in Syropoulos and colleagues.

Syropoulos, S., Mah, A., & Markowitz, E. (2023). Personal legacy: A psychological mechanism for increasing climate action and coping with climate change stressors. *Journal of Mental Health and Climate Change*, 1, 16-29. <http://doi.org/10.5281/zenodo.8320457>

[Response] Thank you for your thoughts.

We based our discussion about generativity on McAdams and St. Aubin (1992) and their conceptualization was in fact extremely broad and included what we now refer to as legacy and responsibility. Specifically, the motivational components of generativity include inner desire to be needed and for symbolic immortality (i.e., lasting agency), which are now commonly defined as legacy motivation. Another motivational component is concern for the next generation which encourage individuals to take responsibility for the future generations. Thus, we do not deviate from the definition that McAdams and St. Aubin (1992) proposed.

Syropoulos et al. (2023, JMHCC) defines generativity as “a person's propensity to engage in acts that promote the well-being of younger generations”, whereas we defined it as the

configuration of psychological features constellated around goals of providing for the next generations, which includes both motives and concerns (legacy and responsibility). Also, it is important to note that Syropoulos defines generativity as a disposition, whereas we treat it as a situational psychological construct.

We respectfully disagree with you that generativity is more biological. Erikson himself discussed Martin Luther and Mahatma Gandhi as exemplars of generativity, and we do not believe that generativity has such a connotation.

Overall, there are several different definitions and approaches towards generativity in the existing literature and we correctly discuss generativity based on the definition offered by McAdams and St. Aubin.

Nevertheless, we can discuss legacy and responsibility without treating them as sub-constructs of generativity. Given the controversial nature of generativity, we now believe it is sensible to discuss legacy and responsibility separately in order to avoid this confusion.

[Revision] First of all, we now propose *three* mechanisms that are unique to intergenerational decision-making: affinity, generativity (responsibility), and legacy.

P3165: “shared group membership between the current and future generations promotes intergenerational cooperation via a set of psychological mechanisms that is unique to intergenerational decision-making^{12,13}: affinity^{3,13–15}, legacy^{14,16}, and generativity (responsibility)^{12,17–19}.”

We have further edited and streamlined the following paragraphs according to this revision.

5. Line 85. Please provide a citation for the definition of legacy.

[Response] It was our oversight we did not provide a reference.

[Revision] We have added relevant references (Wade-Benzoni, 2019).

Study 1

1. I really liked the design of Study 1, and felt that the authors provided a very clear explanation of it. I think Table 1 is a bit confusing. I am not sure what the third column means. So I would recommend revising it for clarity, or simply removing that table as the Figures that follow capture the same information.

[Response] Thank you for your feedback.

[Revision] We have removed Table 1.

2. I think for Table 3, it'd be very useful to add effect sizes (maybe Cohen's *d*) for each comparison. Also, adding ** for $p < .01$, and *** for $p < .001$ would be a nice addition too.

[Response] As we conducted planned contrasts, our preregistered hypothesis testing was based on the logistic regression analysis. As such, the regression coefficients reported in the table themselves are the effect sizes and we thus have offered their confidence intervals. Regarding **, we do not believe it would be a good addition, as the differentiation between p

$< .05$ and $p < .001$ does not convey much information. Also, we do not want readers to base their judgment about effect sizes on those catchy asterisks.

[Revision] No revisions were made.

3. I appreciate the mediation test, but I have a question here. What were the correlations between all variables? It would be good to know, incase there are multi-collinearity issues.

[Response] We have a correlation table in the OSF material (Study 1: isdg_study2.html; Study 2: isdg_JvC.html).

Study 1 (minimal group)

Affinity-Legacy: $r = .55$ [.51, .58]

Affinity-Responsibility: $r = .72$ [.70, .75]

Legacy-Responsibility: $r = .55$ [.51, .58]

Following the correlation analysis, we checked the variance inflation factor (VIF):

Affinity: 2.26

Responsibility: 2.25

Legacy: 1.53

Conventionally, these indices do not present a severe issue of multicollinearity.

To check the robustness of the significant mediation effect via responsibility, we carried out a causal mediation analysis and estimated the average causal mediation effect controlling for the effect of affinity and legacy (n of simulations = 1000, quasi-Byesian Monte Carlo method). The average indirect effect was still significant, $b = 0.59$ [0.32, 1.01], $p < .001$.

Study 2 (Japan vs. China)

Affinity-Legacy: $r = .49$ [.46, .53]

Affinity-Responsibility: $r = .59$ [.56, .62]

Legacy-Responsibility: $r = .50$ [.47, .53]

Following the correlation analysis, we checked the variance inflation factor (VIF):

Affinity: 1.67

Responsibility: 1.68

Legacy: 1.45

Conventionally, these indices do not present a severe issue of multicollinearity.

To check the robustness of the significant mediation effects via affinity, responsibility, and legacy, we carried out causal mediation analyses and estimated the average causal mediation effects controlling for one another as well as reputational concern (n of simulations = 1000, quasi-Byesian Monte Carlo method), which we believe we should have done in the first place in hindsight.

Affinity: The average indirect effect was still significant, $b = 0.10$ [0.02, 0.26], $p = .02$.

Legacy: The average indirect effect was still significant, $b = 0.07$ [0.02, 0.16], $p < .001$.

Responsibility: The average indirect effect was still significant, $b = 0.27$ [0.09, 0.46], $p = .008$.

[Revision] In the result sections, we made notes on the results of causal mediation analyses:
Study 1:

P131273: “We further carried out a causal mediation analysis⁴² (the number of simulations = 1000, quasi-Bayesian Monte Carlo Method), and found a significant average causal mediation effect of responsibility controlling for the effects of affinity and legacy, $b = 0.59$ [0.32, 1.01], $p < .001$.”

P131281: “The causal mediation analysis also revealed the significant average mediation effect of reputational concern, $b = 0.23$ [0.03, 0.85], $p = .03$.”

Study 2:

P171373 “These mediation effects were replicated by causal mediation analyses.”

Study 2

I have the same comments regarding effect sizes, correlations between variables, and p values. Other than these, the design was a nice addition and replication of Study 1 in an intergroup context.

[Response] Please see our response to the previous comment. There, we included statistics for Study 2.

[Revision] Please see our response to the previous comment.

Discussion

I enjoyed the discussion. I think the only addition here would be the authors speculation of what we can do to overcome this outgroup derogation and ingroup favoritism effect in intergenerational decisions. This could spark interesting directions in the field of study.

[Response] Thank you for the positive comment on our discussion and for your thought.

[Revision] We have added a couple of sentences highlighting the potential effect of “universal framing”:

P201445: “In our research, we did not compare the effect of ingroup and outgroup membership labelling against that of universal labeling (i.e., participants know their decision is going to influence everyone in the future, regardless of their group membership). Previous psychological work suggests that an activation of a common superordinate identity improves intergroup relations^{56,57} and such a universal labelling may promote intergenerational cooperation that benefits outgroup members without backlash.”

Minor Comments

1. Rather than saying “more specifically” simply say “specifically”. The word more does not add any additional value.

[Response] Thank you for the catch – we agree with you.

[Revision] We dropped “more” before specifically.

Reviewer #2 (Remarks to the Author):

REVIEW OF COMMSPSYCHOL-25-0028
["Intergenerational Intergroup Cooperation"]

SUMMARY: The paper presents two studies in which participants play ‘intergenerational’ economic games with either ingroup, outgroup, or neutral recipients in later generations. The paper's main finding is that, participants exhibit ingroup favoritism and outgroup derogation also when playing in this intergenerational setting.

EVALUATION: To the best of my knowledge the research design is novel and represents an interesting addition to the literature concerning ingroup favoritism; moreover, the findings appear replicable and robust to group context (minimal vs. national). The main weaknesses of the paper are an unnecessary use of deception and a lack of external validity.

RECOMMENDATION: reject

MAIN REMARKS:

(A) Section 8.07a of the APA’s Ethics Code reads: “Psychologists do not conduct a study involving deception unless they have determined that the use of deceptive techniques is justified by the study's significant prospective scientific, educational, or applied value and that effective nondeceptive alternative procedures are not feasible.” The present study violates this code in at least four ways: (i) participants are led to believe that multiple (!) ‘generations’ of other participants will be involved in the study (see page 39/51 of the experimental instructions), which is not the case. There is only one future generation. It would have been easily feasible to include a third generation in Studies 1 and 2, i.e., to not deceive participants. (ii) Additionally, for Study 2 it is not clear to me if the authors invited participants for the second generation at all. (iii) It remains further unclear if the second generation in both studies were paid and if (iv) they were actually composed of ingroup or outgroup members.

All four mentioned problems indicate unnecessary deception towards the participants. They are severe enough to make me recommend rejection of this paper. I’m aware, though, that this is a strict reading of the Ethics Code and so I’m fine with having my opinion be overruled by a less strict view of the editor and/or a majority of other reviewers.

[Response & Revision] We appreciate your comment on the ethics of research. Let us clarify our instructions and the involvement of deception.

First of all, regarding Study 2, the crowdsourcing platform in Japan did not let us allow for performance-based bonus allotments and we thus could not conduct an incentivized behavioral experiment. We indeed noted this in our manuscript: p161367: “the ISDG was not incentivized”, but we admit that this was not clear enough. In Study 2, we asked them to imagine that they would play the game – thus this study was a vignette study and no deceptions were involved. We did not conduct additional studies to invite future generations. Participants in Study 2 were paid but offered a fixed show-up fee.

To clarify this, we added the following on P161367: “The study was identical to Study 1 except that the study was written in Japanese, the focal intergroup context was Japan (ingroup) vs. China (outgroup), and the ISDG was not incentivized (i.e., we asked

participants to imagine they would complete the ISDG). We note that past research on intergroup cooperation found that the absence (vs. presence) of monetary incentives did not influence results⁴³.”

Regarding Study 1, we told participants that “your choice will determine the options and corresponding rewards available to the next participant” “if you are selected, your decision will influence the next participant’s earnings” Despite that it was implied that the program of the research would include more than two studies, the consequence of participants’ decision was explained without any deceptions.

Following the study, we conducted another study representing the second generation. We randomly picked 10% of the participants in Study 1 and we checked their experimental condition and their decision. We matched the responses from the participants in Study 1 with the experimental condition and the monetary rewards in the follow-up study. For instance, if a selected participant from Study 1 made a sustainable choice in the ingroup condition, we had a participant in the follow-up study from the same minimal group and the payoff structure of the game was identical to that the first-generation participant had. Thus, we do not believe that we deceived participants in Study 1. We revised the manuscript as follows: p81184: “We instructed participants that 10% of participants would receive a bonus payment based on their decision with the conversion rate of £1 = 2000 points and, thus, their decision would influence future participants. To avoid deception, we conducted a follow-up study and recruited 140 additional participants (10% of the participants we recruited for the present study). They completed the ISDG as the second generation of the game and their choices were based on the decisions of the randomly selected 140 participants in the present study. Data from this follow-up study was not analyzed.”

(B) External validity:

I doubt that using an ISDG in an online setting adequately approximates intergenerational dynamics. Its sequence resembles that of a standard dictator game and thus the results might not be interpretable as intergenerational at all: what is missing completely in the design is the present generation’s stake in the outcomes of the future generation (usually: parents would care for their kids’ wellbeing). A refined experimental design would add such a dimension (ideally as an experimental condition) and would further test whether the ‘generation’-story matters at all by removing the deceptive third and later generations to test for any effect of such time horizon.

[Response] Thank you for your thoughtful comment.

First, we do not believe that the ISDG resembles a standard dictator game (DG) for a couple of reasons. A DG is a measurement of prosocial giving/preference and it does not involve a social dilemma: a giver in the game cannot maximize the collective benefit by allocating financial rewards to their receiver, as the money sent does not get doubled or increased by an experimenter. Regarding the intergenerational nature of the game, using multiple experimental sessions is an established way to operationalize intergenerational contexts and this has been used in another commonly used intergenerational economic game, the intergenerational goods game (Hauser et al., 2014 in *Nature*).

Regarding the present generation’s stake in the outcomes of the future generations, the ISDG focuses on two generations that are far apart (i.e., the present generation vs. a future generation that has no overlaps with the present generation). There are studies suggesting that whether or not people are married or have kids do NOT influence intergenerational cooperation (Horn & Kiss, 2020; Tasaki et al., 2023). Specifically, Tasaki et al. (2023) used

the ISDG and found that marital and parental status did not influence decision-making in the game. Thus, we do not believe that such an experimental manipulation would offer more substantial contributions beyond what we have already offered.

[Revision] no revisions were made.

Finally: what type of situation do the authors have in mind with their outgroup condition? Which realistically relevant actions can I take which only affect outgroup members in the future? Climate change, for example, affects both in- and outgroup future generations. Same for most (all?) other environmental dilemmas.

Thus, it remains unclear if the choice of ISDG allows for the results to be generalized to the broad societal implications the paper lists in both introduction and discussion. This is also admitted by the authors themselves. Accordingly, the authors might want to discuss more fitting societal applications for their experimental design.

[Response] Thank you very much for this valuable feedback. We were indeed aware of the limitation of the ISDG and noted this in our discussion, but we agree with you that it is sensible to further expand the discussion and make it clear that our findings are indeed relevant to several real-world contexts.

When we designed our study, we thought about the following contexts:

Cases of China: Chinese industries pollute rivers and polluted water flow into Southeast Asian countries and cause substantial environmental harm in those countries, without significantly affecting that in China. In other words, they are actively harming the local ecosystem in outgroups (i.e., bring substantial immediate future negative outcomes, while benefiting from this action). Similarly, many Chinese companies now have bases in African countries and engage in large-scale mining, leading to the substantial deforestation. This also brings negative future consequences only to outgroups.

Cases of Western developed countries: Many developed countries export electronic waste to developing nations and this has been causing severe pollution and health hazards in outgroups.

These examples highlights that there are indeed many selfish choices that only harm the future of outgroups and their members.

In addition, as we already discussed, past research found that different countries will face different levels of immediate climate change consequences (Lenton et al., 2023). As such, non-sustainable behaviors may not negatively influence one's own country and its people in the next 300 years but the same behaviors may substantially and negatively impact some outgroups and their people. We believe that the ISDG can be thought of as a stylization of such situations too.

[Revision] We have expanded our discussion on the generalizability as follows: p201451, "In addition, our findings are relevant to instances where one country's selfish actions harm the future of other nations, without facing any immediate consequences itself. Historically, developed countries and their industries have implemented a range of policies and practices that substantially harm the ecosystem of other nations (e.g., a dam projects disrupting water

flow to downstream nations and their local eco-system, exporting of electronic wastes to other countries, exploitation of critical minerals from developing countries while causing local environmental destruction, etc.). Consistent outgroup derogation observed in our studies speaks to the prevalence of such cases and our findings on psychological underpinnings may help us design effective interventions. Psychological messaging has been identified as an effective tool to change behavior⁵⁹, and therefore, communications highlighting and triggering legacy motivations and social responsibility for the future may promote intergenerational cooperation for the future of outgroups.

Lastly, we would like to discuss the limitations of the research and future directions. While the ISDG offers a clear measurement of sustainable vs. unsustainable self-serving choices, the intergenerational social dilemma that it represents does not fit all forms of actual intergenerational dilemmas we face. In the ISDG, for instance, it is an individual decision rather than a collective decision that impacts future generations. Thus, our findings may not be generalized to explain decision-making in intergenerational dilemmas in which individuals' decision making cumulatively impacts future generations⁶⁰. While economic games abstract and stylize complex social environments, they are indeed different from actual proenvironmental and sustainable decision making. As such, it is sensible to examine whether our findings are generalizable to a diverse set of actual sustainable decisions.”

Statistics:

The chosen statistical models are to the best of my knowledge applicable for the intended interpretation and the research objectives. The hypotheses are preregistered, and the interpretation of the data is well done. However, given that choosing the cooperative option in every generation would maximize social welfare, efficiency and social welfare concerns of the participants could have been taken into account when eliciting and analyzing the data.

[Response] We agree that individual difference in social welfare concerns may play a role and the examination of individual differences would be one of the directions that future work take our work to. Nevertheless, we did not consider individual differences for a reason. When thinking about policy impact, individual differences are almost always not considered (except for cases of microtargeting: e.g., Tappin et al., 2023) and it is important to identify strategies to promote intergenerational cooperation regardless of individual differences.

[Revision] We had added the following to acknowledge your thought: on p211473, “In our work, we did not consider individual difference factors. On the one hand, our findings suggest the robust effect of group membership on intergenerational cooperation, implying broad policy applicability regardless of individual differences. On the other hand, given the prevalence and effectiveness of microtargeting in political communication^{60,61}, future research could explore individual differences closely, to determine whether and how much group framing influences intergenerational cooperation across individuals.”

MINOR COMMENTS:

The authors should provide additional data to show that the MGI-task induced social identification in the participants. (lines 169 - 172)

[Response & Revision] We have added the following: p111236: “We note that the mean social identification score was 3.31 (SD = 0.09), and it was below the scale midpoint. While the induced social identification was not particularly strong, we found the effect of group

membership on sustainable decision-making. This echoes the past scholarship consistently demonstrating that the strength of social identification does not translate into ingroup favoritism in non-intergenerational social dilemmas^{5,6}.”

The procedure contains a mistake: 10% of the 1400 participants of Study 1 are 140, not as stated 60. If these 60 participants were selected from the pilot data this should be clearly stated. (Lines 188 - 189).

[Response] Thank you for your careful reading. 60 was the number of the participants we collected in the follow-up study for our pilot study (N = 600). We once wrote a paper based on the pilot study and we forgot to replace the number. As we recruited 1400 participants in Study 1, we invited 140 participants in our follow-up study.

[Revision] We replaced 60 with 140.

There are many spelling and grammatical mistakes throughout, for example in lines 161, 166, 167 and 367.

[Response & Revision] Thank you for your careful reading. We have proofread the manuscript carefully and made corrections where appropriate.

Reviewer #2: ingroup/outgroup interactions, large-scale survey

REVIEWERS' COMMENTS:

Reviewer #2 (Remarks to the Author):

REVIEW OF COMMSPSYCHOL-25-0028A
["Intergenerational Intergroup Cooperation"]

SUMMARY: The paper presents two studies in which participants play ‘intergenerational’ economic games with either ingroup, outgroup, or neutral recipients in later generations. The paper's main finding is that participants exhibit ingroup favoritism and outgroup derogation also when playing in this intergenerational setting.

EVALUATION: To the best of my knowledge the research design is novel and represents an interesting addition to the literature concerning ingroup favoritism; moreover, the findings appear replicable and robust to group context (minimal vs. national). The revision addressed most of my concerns and I recommend to accept this contribution after very minor revisions (see below).

RECOMMENDATION: minor revision

[Response] Thank you again for taking your time to review our paper and for helping us improve the manuscript. We have implemented the suggested minor revisions in our revised manuscript.

MAIN REMARKS AFTER REVISION:

Deception:

The revised text improved the clarity of the experimental method. While I still view the instructions and graphics shown to the participants as somewhat deceptive regarding the existence of multiple future generations, in its current form the study's strengths and weaknesses are clear and can thus be assessed by each reader individually. The clarification that Study 2 is an unincentivized vignette is sufficient, as is the added clarification on the payment of the second generation in Study 1. I would however further add a description of how the second generation in Study 1 was matched. This was well formulated in the authors' response to my initial review but is still absent from the paper's text.

[Response and Revision] We revised the section as follows and clarified how we matched the experimental conditions for participants in the follow-up study with the randomly selected participants in Study 1, using an example;

“We instructed participants that 10% of participants would receive a bonus payment based on their decision with the conversion rate of £1 = 2000 points and, thus, their decision would influence future participants. To avoid deception, we conducted a follow-up study and recruited 140 additional participants (10% of the participants we recruited for the present study). They completed the ISDG as the second generation of the game. If a randomly selected participant from Study 1 made a sustainable choice in the ingroup condition, for instance, we had a participant in the follow-up study from the same minimal group play with the identical payoff structure. Data from this follow-up study was not analyzed.”

External Validity:

The provided clarification shows that ISDGs are commonly applied in these settings.

Discussion:

The added examples in the discussion point to fitting real-world equivalents of the stylized decisions of the ISDG.

Welfare Effects:

The revision makes it clear that the mentioned individual welfare considerations were not of interest for the authors' research objectives, the provided clarification suffices. I would have considered a differentiation of these effects as a further improvement of the contribution, but this can also be achieved as part of further examinations in future projects.

[Response] Thank you again for your valuable feedback. We have proofread the manuscript again and corrected typos and grammatical errors.